# NEURAL MECHANISMS OF COGNITIVE FLEXIBILITY: BELIEF UPDATING IN DYNAMIC ENVIRONMENTS WITH SPARSE REWARDS

## ABSTRACT

Humans and animals must develop adaptive strategies to optimize decision-making in dynamic and uncertain environments, often without the benefit of immediate rewards. While existing literature posits that animals use internal "belief" states as the foundation for their decision policy, the mechanism for updating them in a dynamic environment remains unclear. Furthermore, there is no known neural mechanism that can implement belief updates without the need for a reward. To address this gap, we take a multidisciplinary approach that integrates theoretical derivation, training artificial neural networks, and behavioral experiments in rodents to explore potential neural mechanisms of cognitive flexibility.

A belief state is a joint probability distribution over all relevant latent variables of the environment. Updating the joint distributions using only partial observations and marginalizing to obtain estimators is computationally demanding, in particular when latent variables are changing. Moreover, it is nontrivial for a neural network to learn how to implement this complex inference. To tackle these challenges, we introduce a novel change-detection task specifically designed to capture the complexities of partially observed dynamic environments. We formulate a Bayesian theory for sequentially updating joint probabilities and demonstrate that neural networks can accomplish the task near optimally, even in the absence of immediate rewards. We show that the network dynamics mirror the sequential update of the Bayesian latent state estimators. Furthermore, rodents trained on this task show behavior that aligns with our theoretical model and neural network simulations, suggesting that mice utilize dynamic internal state representation and inference to solve this task. Overall, our findings elucidate the computational principles behind flexible cognitive behavior that allows both biological and artificial agents to achieve zero-shot adaptation: modifying their behavior policy to reflect changes in the environment without the need for trial and error.

## 1 INTRODUCTION

If at first you don't succeed, skydiving is not for you. There are scenarios where a trial-and-error approach is simply not an option. In these cases, agents must adapt swiftly to a changing environment, often based only on ambiguous inputs. This necessitates an understanding of how both biological and artificial systems can achieve such adaptability, particularly in the context of Partially Observed Markov Decision Processes (POMDPs), where the complexity introduced by partial observability and sparse reward limits the efficacy of traditional reinforcement learning techniques.

One strategy to circumvent this limitation is to utilize internal "beliefs" (Kaelbling et al., 1998), which are essentially probability distributions over relevant latent states. These belief states are internal representations that facilitate inferences about the environment, enabling agents to make more informed policy decisions. However, belief states are not gracefully implemented by neural networks (Rodriguez et al., 1999). A key question in neuroscience is how brain circuits can overcome this problem (Pouget et al., 2013). The issue becomes even more significant when agents need to rely on their internal beliefs in volatile environments.

Furthermore, most existing literature on belief states has operated under the assumption of stationary environments. This leaves the question of how these internal states could be updated in dynamically changing contexts (Howard & Kahana, 2002). This is a critical gap, especially in cases when understanding that the environment has shifted is crucial before any action can be taken—scenarios where trial-and-error approaches are not only inefficient but also potentially perilous.

To address these challenges, we take an interdisciplinary approach that combines theoretical derivation, neural network simulations, and behavioral experiments in rodents. Our primary focus is on elucidating the computational foundations and neural mechanisms that support cognitive flexibility—the ability to adapt behavior responsively in dynamic settings without the need for an immediate reward.

Our main contributions are as follows:

- We introduce a theoretical framework that elucidates the optimal sequential updates of internal belief states, providing a robust computational basis for decision-making in dynamic environments.

- We demonstrate that neural networks and biological agents rely on a combination of dynamic inference and policy learning. Furthermore, we show, for the first time to our knowledge, that recurrent networks learn sequential Bayesian inference for a POMDP in a volatile environment by means of reinforcement learning while provided only with sparse rewards.

- We show that by basing a policy on dynamic internal belief states, agents can rapidly adapt to a changing environment without acting, effectively achieving zero-shot adaptation.

## 1.1 RELATED WORK

Traditionally, research on associative learning in animals has utilized deterministic cues and overt state representations Vertechi et al.. However, some studies suggest that animals may also employ internal belief states to represent latent features of the environment. Research on internal belief states in animals has mostly focused on reward prediction errors and their implementation by the dopaminergic system (Starkweather et al., 2017; Sarno et al., 2017; Babayan et al., 2018). Other studies confronted the challenges of implementing or learning belief-state representations by neural networks (Rao, 2010; Vértes & Sahani, 2019). A recent study has demonstrated that recurrent neural networks trained under a reinforcement learning paradigm approximate belief states in their neural activity (Hennig et al., 2023). Here, the authors show how recurrent neural networks can predict future rewards directly from observation, utilizing the internal dynamics of the network. Yet, it is unclear how neural circuits learn useful belief representations in more complex environments. As in (Hennig et al., 2023), our work employs Deep Reinforcement Learning (Botvinick et al., 2020) to show that recurrent neural networks learn useful internal belief representations. However, our novelty is considering a dynamic environment with sparse rewards. In our work, internal beliefs change based on observation without the need for an action or a reward. This framework allows agents to adapt rapidly to implicit changes in the environment without changing its synaptic weight and without the need to trial and fail—an important behavior absent in past models.

A different approach for understanding an agent's adaptation to changing environments considers *meta-learning* of latent variables (Wang et al., 2018). Here, neural networks do not learn an internal representation of a latent variable–a belief state. Instead, agents "learn to learn" so they can perform *ad hoc* reinforcement learning of latent variables after training. This line of research has been successful and led to many applications. However, as the name suggests, this framework considers continuous learning and thus relies on acting and error signals. Here, we adapt the same neural architecture as in (Wang et al., 2018) and show it can avoid the trial-and-error paradigm by learning a dynamic inference model. Our approach echoes the ideas of *holistic* reinforcement learning (Radulescu et al., 2019), which argues that neural systems adapt their internal representations to enable better policy learning. In our work, useful representations are neural dynamics that mirror sequential Bayesian inference.

In neuropsychology, *cognitive control* refers to an internal process by which high-level goals modify behavior by providing an appropriate context (Botvinick & Braver, 2015). A large body of work studies how cognitive control shapes decision-making, both at the computational level (Jiang et al., 2014) and the neuro-mechanistic level (Braver et al., 2009). While cognitive control considers context as an internal goal, models often treat it as an explicit signal. In many cases, however, the

context is a hidden property of the environment (Howard & Kahana, 2002; Mante et al., 2013; Remington et al., 2018). In our work, the context is implicit in the observations, and the agent must learn to infer it and dynamically update its internal belief state. The result is an agent that adapts its behavior to changing contexts without explicit control.

This paper is structured as follows: Section 2 introduces the framework and a novel change-detection task designed to probe decision-making in dynamic environments. In Section 3, we derive a Bayesian theory for joint latent variable inference. Our theory serves as a guideline when analyzing trained networks and animal behavior. In Section 4 we train an actor-critic neural architecture on our task. We show that the network's performance compares with a Bayesian agent and use our Bayesian theory to analyze and interpret their dynamics. Finally, in Section 5, we present a physical implementation of our task and results from behavioral experiments with mice. In particular, we provide evidence that mice adapt their behavior based on rapid inference of observations. A short summary and outlook can be found in Section 6. Further details on theoretical derivations, neural network training, and behavioral experiments can be found in the attached appendix.

## 2 PROBLEM SETUP

Our goal is to understand how artificial and biological agents are able to adapt rapidly to changing environments without the need to act and potentially fail. To study this behavior, we introduce a simple *change-detection* task that encapsulates the challenges of acting in an ambiguous and dynamic environment. We begin by defining the state of the world at the time $t$ as $s_t$. The state $s_t$ is pertinent to the task, meaning that the reward is a function of the current state $R_t = R_t(s_t, a_t)$, where $a_t$ is the action chosen at the time $t$. For simplicity, we consider a binary state $s_t = \{0, 1\}$. We refer to $s_t = 1$ as being in a *safe* state and $s_t = 0$ as *unsafe*. The task of the agent is to act ($a_t = 1$) in a safe state ($s_t = 1$), and withhold action ($a_t = 0$) in an unsafe state ($s_t = 0$). Acting in a safe state rewards the agent with $R_t = 1$. As in every POMDP, the states are not directly observed by the agent. Instead, in every time step, the agent receives an observation drawn from a probability function $x_t \sim P(x_t|s_t, \theta_t)$, where $\theta_t$ is a latent variable that is not directly pertinent to the task. Importantly, the environment is dynamic, and both $s_t$ and $\theta_t$ change with time, complicating the inference of the pertinent state $s_t$. Here, $\theta_t$ updates less frequently than the state, which gives it the role of a *context*. Thus, the reward depends only on the state $s_t$, but the context $\theta_t$ is necessary to accurately infer the partially observable state.

In our model, the context $\theta_t$ represents the current uncertainty in the environment. The observable is a binary variable $x_t \in \{0, 1\}$ and at each time step, it is equal to the binary state $s_t$ with probability $1 - \theta_t$ and flipped with probability $\theta_t$. Thus, $x_t = 1$ can be viewed as a *go* signal and $x_t = 0$ is a *nogo* signal, while $\theta_t$ determines how misleading the signals are. For binary contexts $\theta_t \in \{0, 1\}$, observations are equivalent to an XOR Boolean operation, $x_t = \text{XOR}(s_t, \theta_t)$, the simplest example of negative interaction information (Timme et al., 2011). Here, we consider a noisy generalization of the XOR function and allow $0 \leq \theta_t \leq 1$. For brevity, we treat $\theta_t$ as a discrete variable that can have arbitrary-many values within the range; however, it can be easily turned into a continuous variable.

We further break the inherent symmetry of the XOR problem by allowing different levels of uncertainty in the safe and unsafe states. For simplicity, we consider *no uncertainty in the safe state* and the context $\theta_t$ controls the uncertainty in an *unsafe* state. Thus, a *safe* state yields an uninterrupted chain of *go* signals. Conversely, the observations in an *unsafe* state form a Bernoulli distribution with a parameter $\theta_t$. Now, consider an abnormally long series of *go* signals: should the agent infer the state has become safe or that the noise level has increased? In the next section, we will come back to this question.

Finally, the spontaneous dynamics of the environment are governed by the state transition matrix, $\boldsymbol{T}_{ss'} = \delta_{ss'}(1 - 2\lambda) + \lambda$, and the context transition matrix, $\boldsymbol{T}_{\theta\theta'} = \delta_{\theta\theta'}(1 - m\epsilon) + \epsilon$, where $m$ is the number of possible contexts. Thus, the state and context transitions follow Poisson processes with rates $\lambda$ and $\epsilon$, respectively. Importantly, $\epsilon \ll \lambda$, so context switches are less frequent. We note that our theory holds even without the symmetry in the transition matrices. Thus, for example, unsafe states can be longer than safe states or vice versa. The structure of the task is summarized in Figure 1. When the agent acts, the state is changed to an *unsafe* state; if it acts in an *unsafe* state, it receives no reward. Animals have been shown to optimize the rate of reward as opposed to the sum of rewards (Niv et al., 2007) . Maximizing the reward rate requires fast reactions, and the task can

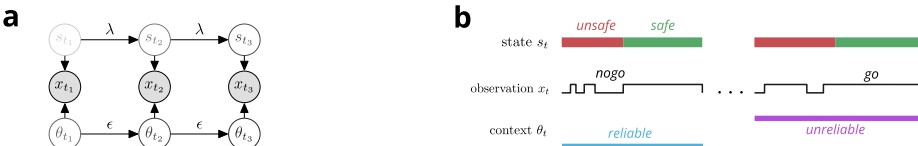

Figure 1: **Change-detection task based on a partially observable Markov decision process (POMDP). (a)** Probabilistic graphical model with negative interaction information $\theta_t \rightarrow x_t \leftarrow s_t$. Transition in the latent state $s_t$ and context $\theta_t$ follow a Poisson renewal process with rates $\lambda$ and $\epsilon$, respectively. **(b)** A sample trajectory based on the model in **a**. While only observing $x_t$, the task is to withhold action in the unsafe state (red) and respond quickly when the state turns safe (green).

be viewed as a change-detection problem. Importantly for us, having no explicit negative rewards allowed us to train mice without subjecting them to actual punishments.

Our framework presents a fundamental difficulty. The agent needs to track both the state and the context in order to choose an appropriate action. This challenge raises two questions: (1) When the statistics of observations change, how can the agent know if it is due to a change in the state or a change in the context? (2) Given that the reward only informs about the pertinent state, can a neural network learn the complex inference of the latent state and context? In the following, we derive a Bayesian theory for the optimal behavior and show that artificial networks learn to approximate the Bayesian solution.

## 3 THEORY FOR OPTIMAL BEHAVIOR WITH INTERNAL BELIEF STATES

In this section, we derive a theory for optimal behavior in the task presented above. We begin by defining an objective and a solution by an optimal observer who has knowledge of the true noise level and only tries to infer the state. Then, we consider the context as a dynamic variable and study the evolution of the belief state, defined by the joint probability of the variables $s_t$ and $\theta_t$.

### 3.1 OPTIMAL POLICY

We start by analyzing the optimal behavior in the simple case of a static environment in which the context does not change, $\theta_t = \theta$ and is known to the agent; we discuss a changing context in the next section. The goal of the agent is to maximize its return. Within a fixed time horizon, this narrows down to maximizing the reward rate, defined by $r = R/T$, where $R$ is the average reward per action, and $T$ is the average time between two actions. Both the average reward and time between actions depend on the policy and the context.

Consider a context without noise, $\theta_t = 0$. In that case, the agent should ideally act immediately upon receiving a *go* signal. However, in a context with nonzero noise, $\theta_t > 0$, there is a $\theta_t$ chance that the *go* signal is misleading and acting will yield no reward. A better policy, in this case, is to wait and accumulate enough evidence (Gold & Shadlen, 2007). Thus, **the policy narrows down to choosing an appropriate waiting time** $\tau$, which we define as the number of *consecutive go* signals ($x_t = 1$) before the agent should act. Clearly, the optimal policy assigns the best waiting time for each context, $\tau_\theta^*$.

Optimizing the reward rate imposes a speed-accuracy trade-off: Waiting longer will increase the chance that the state is safe and increase the average return $R$ per action. On the other hand, longer wait times increase the average time $T$ between two actions. This trade-off implies that when maximizing the reward rate in a given context $r(\tau; \theta)$, there is indeed an optimal waiting time $\tau_\theta^*$ that depends on the context.

The average return when waiting for $\tau$ steps before acting is given by the probability the agent is acting in a safe state, $R = P(s_t = 1 | \theta, \boldsymbol{x}_{t-\tau:t} = 1)$. Here, we used a slicing notation $\boldsymbol{x}_{t_a:t_b} = 1$ to denote that $x_t = 1 \ \forall t_a < t \leq t_b$. The probability of being safe can be written explicitly (see Appendix A.1) as

$$R(\tau; \theta) = P\left(s_t = 1 | \theta, \boldsymbol{x}_{t-\tau:t} = 1\right) = 1 - \frac{b^\tau}{1 - c \sum_{k=0}^{\tau-1} b^k}, \tag{1}$$

where $b \equiv (1 - \lambda)\theta$ is the probability of a misleading *go* (i.e., the probability the state did not change times the probability of it being flipped), and $c \equiv (1 - \lambda)(1 - \theta)$ is the probability of getting observing a *nogo*. Using similar arguments, the average time between actions reads

$$\bar{T}(\tau; \theta) = \frac{\tau b^\tau + \sum_{k=0}^{\tau-1} b^k (\tau \lambda + c(k+1))}{1 - c \sum_{k=0}^{\tau-1} b^k}. \tag{2}$$

Using (1) and (2), one can write an expression for the average reward rate $r(\tau; \theta) = R/\bar{T}$ and find the optimal waiting times $\tau_\theta^*$. Note that $\tau_\theta^*$ is an integer while $\theta$ is a real number. Figure 2a shows samples of the curve $r(\tau; \theta)$ for several values of $\theta$. In each curve, the maximum indicates the optimal waiting time and reward rate for the relevant context. The optimal waiting times directly define the optimal behavior—it defines how exactly an agent should act based on the inputs $\boldsymbol{x}_{\leq t}$. Importantly, from (1), we see that keeping track of the number of consecutive *go* signals, $\tau$, is equivalent to keeping track of the estimated probability that the state is safe

$$\hat{s}_t = \sum_{s_t = \{0,1\}} s_t P(s_t | \theta, \boldsymbol{x}_{\leq t}) = P(s_t = 1 | \theta, \boldsymbol{x}_{t-\tau:t} = 1). \tag{3}$$

Here, the notation $\boldsymbol{x}_{\leq t}$ denotes the current and all previous observations. Thus, using the number of consecutive *go*s as a proxy, we have calculated the **optimal policy** $\pi^\star(a_t | \hat{s}_t, \theta)$. Simply put, the policy is to act if and only if the estimate of the state is equal to or larger than the threshold, which we define as $\hat{s}_\theta^\star = P\left(s_t = 1 | \theta, \boldsymbol{x}_{t-\tau_\theta^*:t} = 1\right)$, obtained by plugging $\tau_\theta^*$ into (3).

## 3.2 BELIEF STATES

In a dynamic environment, where both the state and the context can change, the agent must keep track of the latent variables $s_t$ and $\theta_t$. As we have seen above, the current state can be estimated by counting the number of consecutive go signals $\tau$. We note that the exact way in which the estimator $\hat{s}_t$ changes with each additional go signal depends on the current context $\theta_t$.

Within the unsafe state, $s_t = 0$, the context can be inferred as a simple running average over recent inputs $\hat{\theta} = \frac{1}{n} \sum_{t-n}^t x_t$. The longer the window for averaging $n$, the more accurate the estimate. However, since the state can change, the agent needs to know whether to attribute a string of consecutive go cues to a noise fluctuation or a state change. The same string of $\tau$ go cues should change $\hat{s}$ or $\hat{\theta}$ with some probability that depends on the current estimates $\hat{s}$ and $\hat{\theta}$.

In this case, the belief state is given by the conditional joint probability $P(s_t, \theta_t | \boldsymbol{x}_{\leq t})$, denoting the probability of finding $s_t$ and $\theta_t$ given the past observation. Here, we note the history dependence of the joint probability, conditioned on all observations $\boldsymbol{x}_{\leq t}$. Importantly, the conditioned probability does not factorize because of the interaction between state and context when generating observations. From the belief states, one can obtain the estimators for the current state and context using the marginal probabilities

$$\hat{s}_t = \sum_{s_t, \theta_t} s_t P(s_t, \theta_t | \boldsymbol{x}_{\leq t}), \quad \text{and} \quad \hat{\theta}_t = \sum_{s_t, \theta_t} \theta_t P(s_t, \theta_t | \boldsymbol{x}_{\leq t}). \tag{4}$$

**Sequential updates of the belief states.** To see how the belief $P(s_t, \theta_t | \boldsymbol{x}_{\leq t})$ updates with new observations, we extend the *Chapman-Kolmogorov* equation (Gardiner et al., 1985) for joint conditional probabilities to explicitly write how the joint distribution is updated with new observations. The update rule is given by

$$
\begin{aligned}
P(s_t, \theta_t | \boldsymbol{x}_{\leq t}) &\propto P(s_t, \theta_t, x_t | \boldsymbol{x}_{\leq t-1}) \\
&= P(x_t | s_t, \theta_t, \boldsymbol{x}_{\leq t-1}) P(s_t, \theta_t | \boldsymbol{x}_{\leq t-1}) \\
&= P(x_t | s_t, \theta_t) \sum_{s_{t-1}} \boldsymbol{T}_{s_t s_{t-1}} \sum_{\theta_{t-1}} \boldsymbol{T}_{\theta_t \theta_{t-1}} P(s_{t-1}, \theta_{t-1} | \boldsymbol{x}_{\leq t-1}).
\end{aligned}
\tag{5}
$$

The proportionality constant can readily be turned into an equality by normalizing the updated equation. On the left-hand side of (5) we have the joint probability of the state and context given all past observations; on the right-hand side, we have the belief state at the previous time states, weighted by the transitioning probability and the current observation. For initial conditions, we assume $s_0 = 0$ and $\theta_0$ is distributed according to the equilibrium of the transition matrix, though at large $t$ the system is agnostic towards the initial states. Together, equations (4) and (5) describe how the Bayesian estimation of the state and context gets updated with observations.

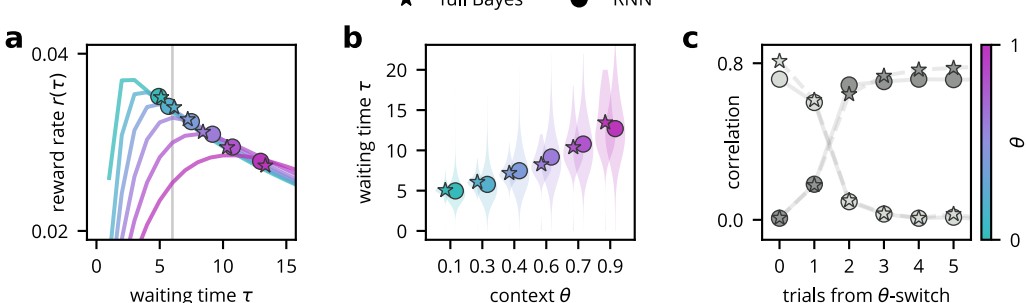

Figure 2: **Optimal behavior.** The behavior in the task is defined by how many time steps the agent should wait before it determines the state is safe. **(a)** Reward rate curves as a function of the waiting time $\tau$ from the last *nogo* signal, given by $r(\tau; \theta) = R/\bar{T}$. Colors indicate the context values $\theta$. Both Bayesian estimators and RNN performance are suboptimal because of errors in context estimation, but their performance is similar. The vertical line denotes the optimal waiting time averaged across all contexts. **(b)** Context-dependent waiting time distributions. Network and Bayesian agents show similar mean waiting times. **(c)** Adaptation behavior after a context switch. On the $x$-axis is the number of trials (actions) from a context switch (0 being the last trial of the previous context), and on the $y$-axis is the Pearson correlation of the agent's waiting time with optimal waiting times of the current (dark shade) and previous (light shade) contexts. Importantly, the behavior shows a nonzero correlation even before the first reward in a new context.

**Policy.**    Finally, the optimal action is determined using the current Bayesian *estimates* of the state and context but is otherwise identical to the policy calculated in the previous section:

$$\pi^{\star}(a_t|\hat{s}_t, \hat{\theta}_t) = \begin{cases} 1 \text{ (act)} & \hat{s}_t \geq \hat{s}^{\star}(\hat{\theta}_t) \\ 0 \text{ (don't act)} & \text{otherwise} \end{cases}. \tag{6}$$

To see how the model behaves, we tested it on our task. In Figure 2a we see that the model behaves close to optimally in each context, suggesting a good inference of the underlying $\theta_t$. Indeed, the waiting time distribution was dependent on the context, as seen in Figure 2b. In Figure 2c. we show the correlation in waiting times between the model and the optimal waiting times of an ideal observer who has access to the true $\theta_t$. Importantly, the inference of the new context is rapid, and behavior is adapted already within the first trial. These results are not surprising, as they rely on input inference and not the reward to detect changes. However, these theoretical results serve as a baseline when we study networks' and rodents' behaviors in the following sections.

We summarize the two main features of our model. First, the internal states' update does not require the system to act. Thus, the agent can update its internal state rapidly as the environment changes. Second, the optimal policy depends on the point estimators and not the full joint probability distribution; this is because the system can infer the state of the world from the input and *does not require exploration*. As a result, the optimal policy is deterministic and requires only the best estimate of the latent variables (Sutton & Barto, 2018).

## 4  NEURAL NETWORKS MIRROR BELIEF-STATE ESTIMATORS UPDATE

In the previous section, we have shown that the optimal policy depends only on the Bayesian estimates of the latent variables conditioned on past and current observations. Calculating the estimators requires keeping a representation of the full joint probability function $P(s_t, \theta_t|\boldsymbol{x}_{\leq t})$ and implementing the complex dynamics defined by (5). Holding track of the full probability becomes prohibitively costly as the latent dimensionality increases. In this section, we show that neural networks trained on our task represent the Bayesian estimators $\hat{s}_t$ and $\hat{\theta}_t$, and mirror their sequential updates conditioned on the current observation.

**Neural architecture and training.**    We trained artificial neural networks to solve the task using a deep reinforcement learning algorithm (Botvinick et al., 2020). Our neural architecture utilized

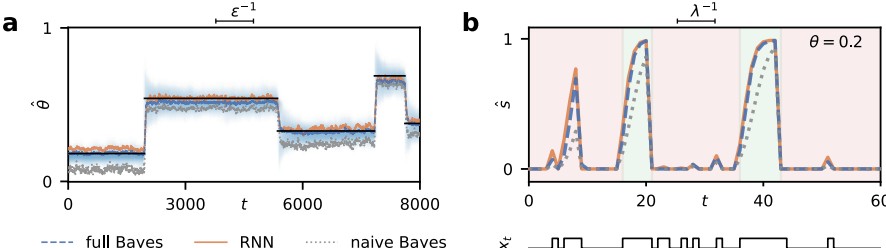

Figure 3: **Network activity encodes Bayesian estimation of the state and context. (a)** Using a linear readout from network activity, we decode the estimate $\hat{\theta}_t^{\mathrm{net}}$ (solid orange) of the current context $\theta_t$ (solid black). As a point of comparison, the naive Bayesian model (dotted grey) is noticeably worse. Shown is an average of 45 realizations of $s_t$ and $x_t$ trajectories. **(b)** Network decoded belief $\hat{s}_t$ (solid orange) together with the full Bayesian estimator $\hat{s}_t^{\mathrm{net}}$ (dashed blue) and the naive model (dotted grey) for a sample $x_t$ trajectory (bottom). Overall, our analysis shows that the network closely follows the Bayesian estimators produced by the joint belief state, updated using (5). The scales on top denote the context and state transition time scales.

an actor-critic framework previously used on related sequential decision-making tasks (Wang et al., 2018). The input to the networks was the observations $x$, and the output at any given time was whether the network chose to act or not. The network received a reward whenever it acted in a safe state. We report here the results of the training of an LSTM architecture (see Appendix A).

**Neural networks represent Bayesian estimators.** First, we asked whether the network activity represents the estimators. For that, we trained a linear decoder to read out the *true* latent variables. We used logistic regression to classify the true state $s_t \in \{0, 1\}$ and the resulting readout probability for $\hat{s}_t^{\mathrm{net}}$. To obtain the estimation of the continuous context variable, we used linear regression on the *true* context $\theta_t \in (0, 1)$ and obtained a set of readout weights for $\hat{\theta}_t^{\mathrm{net}}$. For the analysis, we used data unseen by the regression. We emphasize that the readout weights are not time or context-dependent and are calculated once for the network. To assess how well the network readout performs, we measured the root mean square error (RMSE) across different times and inputs. While regressing to the true value, our readout showed a closer fit to the Bayesian estimators (Figure 3). In particular, the context estimation $\hat{\theta}_t^{\mathrm{net}}$ had a better fit with the Bayesian estimate $\hat{\theta}_t$ (RMSE=0.062) compared with the true context $\theta_t$ (RMSE=0.098).

To further support the claim that the network approximates the joint probability $P(\hat{s}_t, \hat{\theta}_t | \boldsymbol{x}_{\leq t})$, we compare the result to a *naive* Bayesian estimate, in which the agent is unaware of the connection between the state and the context and estimates two independent beliefs, $P(\hat{s}_t | \boldsymbol{x}_{\leq t})$ and $P(\hat{\theta}_t | \boldsymbol{x}_{\leq t})$. This corresponds to the vertical line in Figure 2a. In the naive model, the state inference is context-independent and is bound to perform worse (dotted line in Figure 3). Nevertheless, it is a useful baseline that suggests the network's internal belief is based on the joint probability. The derivation of the naive model's dynamic is detailed in Appendix A.1.2.

In addition, in all examples we show here, we use context values $\theta_t$ that were not used to train the recurrent networks. This shows that the networks have learned to generalize correctly to unseen contexts in the range $0 < \theta_t < 1$ and did not just memorize a set of parameters.

**Network dynamics mirrors the sequential dynamics of the Bayesian estimators.** Recurrent neural networks are high-dimensional, nonlinear dynamical systems, which are *a priori* well-suited for approximating sequential updating. However, it is difficult to determine whether the networks actually learned the complex dynamical landscape underlying the Bayesian inference or that they follow more trivial trajectories. To test that, we examine their response to different observations $x_t$ under different conditions by comparing the linear readouts $\hat{s}_t^{\mathrm{net}}$ and $\hat{\theta}_t^{\mathrm{net}}$ to the Bayesian theory.

In Figure 4a, we look at the update of the belief $\hat{s}_t$ to a series of consecutive *go* signals in the unsafe state. Each curve represents the average over different realizations of *past* observations. The graph shows how the probability of being safe increases with $t$ consecutive *go* signals, reflecting the integrator dynamics of the optimal policy derived in Section 3. Furthermore, the belief collapses

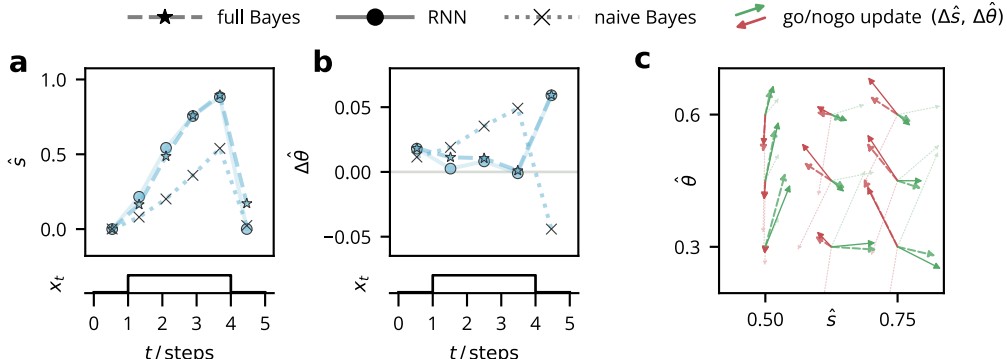

Figure 4: **Network dynamics implement sequential updating of the joint Bayesian estimators.**
**(a)** State estimate of the network $\hat{s}_t^{\mathrm{net}}$ (circle), Bayesian $\hat{s}_t$ (star), and naive model (cross) as a function of consecutive *go*s in the unsafe state. A single *nogo* signal collapses the state estimate, as the state is evidently unsafe. Network estimates follow the Bayesian updates closely. **(b)** Update of the context estimate $\Delta\hat{\theta}_t$, for the network (circle), Bayesian (star), and naive model (cross). A sharp increase in context estimation after a *nogo* is a hallmark of the joint probability update. **(c)** Update dynamics of estimators as a function of current state $\hat{s}_t$ and context $\hat{\theta}_t$ estimates. Arrows denote updates for *go* (green) and *nogo* (red) observations with the same linestyles as in **(a)** and **(b)**. Arrow lengths are scaled to compensate for the exponentially saturating belief in $\hat{s}$. Solid, dashed, and faded denote network, full Bayesian, and naive models, respectively. Overall, the network activity approximates the dynamics of the Bayesian estimators $\hat{s}_t$ and $\hat{\theta}_t$.

immediately with a single *nogo* signal, which indicates the state is unsafe. For comparison, we show the naive inference, which doesn't reflect the state-context coupling. We observe that the network follows the dynamics of the joint-distribution estimators.

To further analyze the dynamics, we look at the updates to the estimators in each step, $\Delta\hat{s}_t^{\mathrm{net}} = \hat{s}_t^{\mathrm{net}} - \hat{s}_{t-1}^{\mathrm{net}}$ and $\Delta\hat{\theta}_t^{\mathrm{net}} = \hat{\theta}_t^{\mathrm{net}} - \hat{\theta}_{t-1}^{\mathrm{net}}$, and compare them to the update obtained by the Bayesian model $\Delta\hat{s}_t$ and $\Delta\hat{\theta}_t$ from (5). In Figure 4b, the update of the context estimate, $\Delta\hat{\theta}_t$ is plotted as a function of consecutive *go* signals followed by a *nogo*, again, averaged over different histories. A hallmark of the joint probability update is that a single observation can sharply increase the context estimate, because recent positive observations can now be explained by high unreliability $\theta_t$. This contrasts naive inference, in which the estimation of $\hat{\theta}_t$ always increases in response to a *go* signal, and decreases following a *nogo*. Again, networks follow estimators of the joint probability.

Finally, the updates of the estimators to *go* and *nogo* signals in the phase space of $\hat{s}_t$ and $\hat{\theta}_t$ are depicted in Figure 4c. First, the update dynamics are complex; *go* and *nogo* signals result in different updates that depend on the current estimate. Second, the network dynamics approximate the dynamics of the Bayesian theory we derived in Section 3; this is emphasized when contrasting the naive estimator. Third, looking at the update arrows in Figure 4c, one can trace out an approximate trajectory in the state space for either observation. If we had added uncertainty to the safe state, we could have taken the limit where the discrete update becomes a continuous flow. This analysis is beyond the scope of this work.

Overall, our analysis shows that recurrent dynamics implement the sequential update to the estimators over the joint Bayesian belief state described by (5).

## 5 MICE EXPERIMENTS SHOW RAPID COGNITIVE FLEXIBILITY

To test whether our paradigm can capture animal behavior, we trained adult mice on a physical instantiation of our task. We chose an auditory stimulus modality, as it allows for the precise control needed for the task, and mice are proficient in auditory tasks. In the experiments, *go* signals ($x_t = 1$) were presented as a tone (an accord with five frequencies) and *nogo* signals ($x_t = 0$) were silent. Each step was 0.2 seconds long and gapless, so consecutive *go* signals were presented as

a continuous tone. Mice were water-deprived, and a successful action rewarded the animal with a drop of water.

Generally, the mice struggled to learn the task, which required us to design suitable adjustments. First, after an action, an inter-trial interval (ITI) was introduced. During the ITI, the mice were exposed to the unsafe state's observations but were not punished for licking. This allowed us to overcome the tendency of mice to lick several times when allowed. For simplicity, the ITI can be viewed as prolonged unsafe states. To make sure the effects on the behavior are minimal, we derive the state inference in the presence of ITIs in Appendix A. The second modification was extending the safe period to allow the mice to learn the task.

Overall, the unsafe state's duration time scale was 2 seconds ($\lambda = 0.5\,\mathrm{Hz}$), ITIs were 2 seconds long, and the safe state was up to 7 seconds long. We trained 11 mice for 3-4 weeks, after which we collected the behavioral data in 10 sessions from each mouse. Each session held at least 200 trials.

In general, mice were more impatient than artificial networks and showed sub-optimal performance. This can likely be attributed to the effects of perception, motor delays, and environmental factors that influence animal behavior. However, the statistical analysis of the data, together with our model, reveals systematic behavior and suggests that mice may use dynamic internal belief as a mechanism for cognitive flexibility in this task.

First, we see that mice modulate their behavior according to the current context. This can be seen by the change in average waiting time in Figure 5a (compare with Figure 2b). This finding suggests that the mice learn the underlying structure of the task. Second, the mice's response is modulated on the first action, as can be seen by the correlation between the recorded and optimal waiting time of the current and previous context after the switch (Figure 5b, and compare with Figure 2c). This finding suggests that the mice infer the latent states of the task based on the auditory input, and adjust their behavior rapidly without requiring feedback from the environment. In the future, we plan to analyze neural data to test this hypothesis.

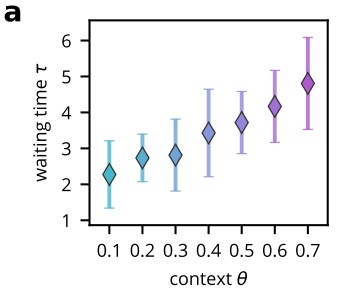 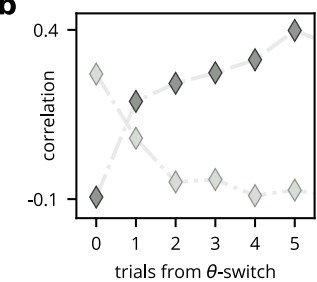

Figure 5: **Mice exhibit rapid context adaptation.** Behavioral (waiting time) analysis for mice performing the task. **(a)** Same as Figure 2b , only with mice data. **(b)** Same as Figure 2c, only with mice data.

## 6 SUMMARY

We have presented an interdisciplinary investigation into the neural mechanisms underpinning cognitive flexibility, specifically the ability of biological and artificial agents to update belief states in dynamic environments without relying on a trial-and-error approach. We introduced a novel change-detection task designed to explore the computational challenges of solving POMDPs in dynamic environments. The design of this task allowed for a multimodal approach: (1) We derived a theory for sequential Bayesian inference of joint probabilities; (2) we analyzed the dynamics of artificial neural networks solving the task and showed it mirrors the sequential update of the Bayesian estimators; and (3) we showed that mice are able to adapt their behavior without a reward, resembling the Bayesian and network policies..

In conclusion, our research offers a stepping-stone towards understanding the complex behavior of intelligent agents. The ability of recurrent neural networks to leverage their internal dynamics for nontrivial world inference underscores the potency of internal belief states underlying cognitive processes. Our finding that neural networks can implement nontrivial inference using only experiential learning makes this a potential mechanism for adaptive behavior. We believe it can pave the way for the development of more sophisticated models that can further refine our understanding of cognitive flexibility.

REPRODUCIBILITY STATEMENT

We provide the code for our simulations, with scripts to reproduce the figures. This includes an archived version of the network that we trained using Pytorch, as well as scripts to replicate the training. Behavioral data used for Figure 5 will be available upon request with the publication of this work.

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
