# OpenReview forum: "Neural mechanisms of cognitive flexibility: Belief updating in dynamic environments with sparse rewards"
_ICLR.cc/2024/Conference — ICLR 2024 Conference Withdrawn Submission_

### Official Review · Reviewer_JMWQ · 2023-10-27

**Soundness:** 3 good
**Presentation:** 4 excellent
**Contribution:** 1 poor
**Rating:** 3
**Confidence:** 4

**Summary:**

In this work, the Authors investigate the ability of LSTMs and mice to update belief states in dynamic environments by relying on observations instead of rewards. To this end, they have implemented an HMM task, for which they offer a theoretical solution, an LSTM-based simulation, and a mouse experiment. Based on experimental observations, they conclude that LSTM activity and mouse behavior in their task resemble the updates of the Bayesian estimator in their theory, potentially offering insights into the underpinnings of the rapid adaptivity of real-world agents.

**Strengths:**

The writing is super clear, making it easy to follow the Authors’ argument.

The general composition of the research and the paper is exactly what’s expected from a good paper in neuroscience as it combines theory, simulation, and an animal experiment.

The coverage of related literature is decent, mentioning many of the foundational papers that come to mind after reading the abstract.

The goal of this work is worthy and timely, as an increasing number of groups get interested in theoretical/neural underpinning for the hypothesis of the Bayesian mind.

Newly collected data is a plus.

**Weaknesses:**

Considered in isolation, there would be no doubt that this work is a great piece of research. It seems, however, to be closely related to many other works, both cited in this paper or not, which requires the gauging of the reported results by those in prior literature. This gauging would necessitate a highly detailed comparison between the models/results in prior work vs here, which I wasn’t able to gather from this text alone. Details below.

I think the easiest way to go here is by the stated contributions of this paper. On Page 2, it’s stated that

(1)	[They] introduce a theoretical framework that elucidates the optimal sequential updates of internal belief states;

(2)	[They] demonstrate that neural networks and biological agents rely on a combination of dynamic inference and policy learning.

(3)	[They] show, for the first time to [their] knowledge, that [RNNs] learn sequential Bayesian inference for a POMDP in a volatile environment by the means of [RL] while provided only sparse rewards.

(4)	[They] show that, by basing a policy on dynamic internal belief states, agents can rapidly adapt to a changing environment without acting.

On Page 4, they continue to a result that:

(5)	The policy narrows down to choosing an appropriate waiting time.

And then, on Page 7, they conclude that:

(6)	The optimal policy depends on the point estimators and not the full joint probability.

In the grand conclusion, they summarize that:

(7)	The dynamics of [ANNs] […] mirrors the sequential updates of the Bayesian estimators.

(8)	Mice were able to adapt their behavior without a reward, resembling the Bayesian and network policies.

Arguably, (1), (3), and (7) are highly similar to a recent-ish work form Sam Gershman’s lab, cited in this paper [Hennig et al, PLOS Comp Bio 2023]; (2) is similar to a work from Matt Botvinick’s group, cited in this paper [Wang et al, Nat Neurosci 2018]; (4), (5), (6), and (8) are similar to an older work from Jon Cohen’s group [Yu & Cohen, N[eur]IPS 2008], not cited in this paper.
To compare with Sam Gershman’s paper, the Authors say that their task is more complex, however, the complexity is a subjective measure. Overall, this leads me to the main question (below):

**Questions:**

-Could you please describe (ideally – in detail) how your work differs from running Angela Yu’s kind of HMM on Jay Hennig’s kind of inference model using Jane Wang’s kind of architecture, why does using observations instead of rewards make a difference for Bayesian inference, and what new insights does the work offer compared to the aforementioned studies? Distinguishing current results from the results in past literature is highly important for the appropriate evaluation of the current work, and will help the Reviewers and the Area Chairs to provide a fair assessment. Until then, the available information deems insufficient to make a decision regarding the recommendation of this work for acceptance to ICLR.

Minor questions:

-There’s a really interesting point in the discussion of the behavioral data as to that the animals did underperform compared to the past studies. I was wondering if this could be explained by one of the existing theoretical and/or experimental observations, available in literature. For example, [Pisupati et al, eLife 2021] suggest that lapses in decision-making may be caused by the exploration; [Ma & Hermundstad, biorxiv 2022] argues that Bayes-optimal inference may be approximated by heuristics (that could potentially be easier to compute), and [Yu & Cohen, N[eur]IPS 2008] argues theoretically, followed up by [Shuvaev & Starosta et al, NeurIPS 2020] arguing experimentally that lapses may result from hardwired generalist rules applicable beyond the experimental settings. I wonder if any of these would apply to your data.

-Did you consider any alternative explanations for the observed neural activity in the LSTMs? Having and rejecting those would strengthen an argument about the consistency of the observed neural code with Bayesian inference.

Minor remarks:

-Page 1: “How brain circuits can overcome [the implementations of belief states by neural networks]” is, perhaps, not a “key question in neuroscience” but, at a minimum, one of these.

-Page 2: [Botvinik et al, 2020] is, perhaps, not *the* reference for the (entire field of) deep RL. If it was intended to reference a particular implementation used in this work, I’d suggest making it more clear.

---

### Official Review · Reviewer_pqJH · 2023-10-31

**Soundness:** 1 poor
**Presentation:** 1 poor
**Contribution:** 2 fair
**Rating:** 3
**Confidence:** 4

**Summary:**

This paper introduces a multi-disciplinary work across Statistics, Machine Learning and Behavioral Biology on decision making. The paper focuses on the environments whose reward is not immediate or quite sparse (only sense it when certain state is achieved.)

**A POMDP problem** of 2 states, 2 observations and 2 actions with only non-negative rewards is introduced and is constructed in real world for rodents. Theoretical results are provided, together with performance of neural networks and real animals.

- Action strategies are given theoretically, for fixed observation noise first. Then by calculating a belief of noise and state (posterior based on observation and experiments of action-result history). Combining these two parts completes the theoretical solution.

- A deep RL solution is given, following an actor-critic framework and LSTM architecture. A comparison on behavior change and belief prediction between the LSTM and Bayesian belief updates (above) is provided, showing certain similairty.

- The task is implemented suitable for rodents. If a water-deprived mouse reads the signal and infer the states correctly, a drop of water is given as reward. The experiment shows an reasonable waiting time (strategy given a fixed context) rapid context adaptation of mice.

**Strengths:**

The idea of a parallel experiment and coincidence across theory - deep RL - animal is good. And result shows certain coincidences.

**Weaknesses:**

- The statement of the basic problem (the POMDP environment) is hard to understand. I could find certain self-conflict parts.

- The environment is too simple, for theory and neural networks, after all.

- The method used in theoretical calculation part is problematic.

- Lack of certain data-theoretical comparison between animal experiment and Theory / RL results.

**Questions:**

The statement of setup is unclear and sometimes self-conflict, so I cannot guarantee the methods and results are correct. I am looking forward to receiving a clearer and self-consistent explanation about the following problems.

## Problems and Typos

1. Section 2, the end of line 6, I guess it should be $s_t\\in\\{0,1\\}$ instead of $s_t=\\{0,1\\}$

2. Section 3.1 Paragraph 2, line 2, "there is a $\\theta_t$ chance that the go signal is misleading and acting will yield no reward".
The probability is for the misleading *go* signal among all *go* signal cases. Thus it depends on the probability of $s_t=1$.
In the stable state of the Markov chain for symmetric $\\mathcal{T}$ where $P(s_t=1)=0.5$, the value
should be $\\frac{\\theta_t}{1+\\theta_t}$ instead.

3. Section 3.1 Paragraph 2. The statement contradicts with previous setup of Transition function $\\mathbf{T}$.
    - The POMDP setup of the problem defines transferable states, i.e., $s$ can change from 0 to 1, also can change from 1 to 0, following the symmetric transition function $T_{ss'}=(1-2\\lambda)\\delta_{ss'}+\\lambda$.
    - However, "the policy narrows down to choosing an appropriate waiting time $\\tau$, which we define as the number of consecutive go signals($x_t=1$) before the agent should act." is a result that the state $s_t=1$ **once achieved would not change back to** $0$ (If so, the *go* signal $x_t=1$ may change back to 0 after a while)!

    - So, what is the real setup of the environment? **In the following checks, I follow the **one-way** setup of no transition from s=1 to s=0, since otherwise, I should just stop here and conclude that the waiting time method is wrong.**

4. Appendix A for probability distribution after $\tau$ consecutive *go* signal. Based on **one-way** setup, the probability is simply $1-(b/c)^{\tau}$ using the symbols in the paper, and the optimal waiting time depends on the current round.
    - After all, $P_{s=0}=b^\tau+\sum b^kc P_{s=0}$ is problematic, as the $P_{s=0}$ is not stable in the procedure, they cannot be assumed the same.

5. The mice data figure 5b is not explained clear enough. What do the two colors mean?

---

### Official Review · Reviewer_qv9Y · 2023-11-01

**Soundness:** 3 good
**Presentation:** 4 excellent
**Contribution:** 1 poor
**Rating:** 3
**Confidence:** 3

**Summary:**

The paper proposes a new simple change detection task that can be used in rats (and models) to investigate the interplay between state inference and reward-based decision making in a changing environment. An approximately optimal policy is derived for the task, and approximated by a neural network. Furthermore, rats are shown to exhibit qualitatively similar behavior to the models after appropriate modification and training.

**Strengths:**

This is a clear and well-written paper - the flow is easy to follow, notation is crisp, and it does well to reduce its core question to toy task that can be investigated exhaustively.

**Weaknesses:**

I have to primary concerns: the novelty of the contribution, and the overall magnitude / impact for the ICLR audience.

## Novelty
The paper seems to miss a notable body of work on optimal change detection and decision making in changing environments. One specific line of work is from the Gold/Kable labs, notably Glaze et al. 2015 doi:10.7554/eLife.08825 and related papers. Glaze et al. have a normative model of evidence accumulation in unstable environment and related work from Josh Gold's lab (with various lead authors) works on other aspects of change point problems.

A second line of work is from Josic and Kilpatrick, e.g. Radillo et al. 2017 doi:10.1162/NECO_a_00957, Kilpatrick et al. 2019 doi:10.1016/j.conb.2019.06.006, Barendregt et al. 2019 doi:10.1007/s10827-019-00733-5. Radillo & colleagues seem to have a more general case of the change detection task derived, including an approximation by a neural network, and more recent work builds on it further.

I think the present work needs to be contextualized relative to this past work. Perhaps I am misinterpreting the connection, or perhaps the simplified task is more amenable to analysis or for training rodents, or has some other benefits, etc, but this needs to be made explicit and defended. On a technical front it's also not obvious that the proposed policy extension to the dynamic case (by tracking a running average over recent inputs) has appropriate normative guarantees (cf. the work cited above, which makes its approximations explicit).

## Overall significance and impact to the audience
Much as I love to see theoretical and empirical neuroscience work appear in ML venues, I'm not sure that this contribution is sufficiently interesting to the broader ICLR audience. The toy task as proposed may be great for building intuition but a model that solves it is not necessarily impactful, and the empirical results in rats do not carry an ML paper either. I can imagine more general analysis being more impactful (though that may also make this even more of a MathPsych paper), or a stronger empirical result with more sophisticated tasks/models, etc. But I'm not sure I see enough here, especially considering the prior work outlined above.

Minor comments:
* In Section 1.1 Vertechi et al. cite looks incorrectly typeset (\citet instead of \citep?)
* "Getting observing a nogo" on the top of p5
* Fig 4c is not discussed in the caption, though it is discussed in the text.
* I would consider adding some statement regarding appropriate use of animal subjects to the main text rather than the supplement only, if space permits.

**Questions:**

* The RNN seems conservative (always wait longer than the optimum would suggest). Is this bias driven by some aspect of the setup or something else we can understand about the network's behavior or training?

---

### Official Review · Reviewer_mk4V · 2023-11-01

**Soundness:** 3 good
**Presentation:** 3 good
**Contribution:** 2 fair
**Rating:** 5
**Confidence:** 4

**Summary:**

The paper explores the neural mechanisms underlying cognitive flexibility, particularly focusing on belief updating in dynamic environments with sparse rewards. A simple change-detection task is introduced, and Bayesian belief update equations derived for state and context. The results demonstrate that artificial neural activity closely approximates Bayesian estimators. Interestingly, rodent behavioural experiments are included that show mice may utilise dynamic internal beliefs for decision-making, albeit in a sub-optimal manner compared to metal-learnt LSTMs.

**Strengths:**

Overall, I find the paper interesting - although bit contrived in the task being evaluated:

1.	The multidisciplinary approach provides a unique way of considering the computations that might underwrite cognitive control.
2.	The empirical validation using rodents and comparison with the LSTM formulation was useful in considering how biologically plausible the learning process was. From my perspective, it makes a paper interesting for both neuroscience, cognitive science, and machine learning audiences.
3.	Simplicity of the task makes it straightforward to validate the results from the LSTM formulation with respect to Bayesian latest state estimations.  Additionally, an explicit evaluation of whether a particular type of recurrent network architecture can learn sequential Bayesian inference in a volatile environment by means of reinforcement learning while provided only with sparse rewards.

**Weaknesses:**

1.	The paper introduces a simple change-detection task to probe decision-making in dynamic environments. While this is a useful for face-validation, it also limits the generalisability of the findings to other types of tasks or environments.
2.	The paper acknowledges that calculating Bayesian estimators requires keeping a representation of the full joint probability function, which becomes prohibitively costly as the latent dimensionality increases. It would be useful to have an assessment of whether the rodent experiments are not aligned because of computational complexity associate with evaluating these estimators and perhaps, a simpler approximation is sufficient instead?
3.	There was no reference to how this formulation compares to variational /amortised inference approaches which are also formulate belief updating in dynamic settings e.g., hierarchical Bayesian models (e.g., hierarchical gaussian filtering, deep active inference, meta-level MDPs etc). This is particularly relevant because biological agents don’t follow the Bayes-optimal policy.

**Questions:**

1.	How does behaviour follow optimal policy in more naturalistic environments with multiple stimulus?
2.	If there are multiple different transitions over different latent states (s_1, s_2…) are evolving at different timescales. How would this change the derived reward rate? And would we expect the simple RNNs to be able to model period cycles.
3.	\theta_t + s_t are both pertinent to the task inference. Would it be possible to do an ablation analysis where both are evolving but inference is only being carried out given the $s_t$

Minor:
1.	The first sentence in the related work section has a typing error wrt citation.